# DRIMA: DIFFERENTIAL REWARD INTERACTION FOR COOPERATIVE MULTI-AGENT REINFORCEMENT LEARNING

## ABSTRACT

Multi-agent reinforcement learning (MARL) owning to its potent capabilities in complex systems has gained remarkable research attention nowadays, in which collaborative decision-making and control for multi-agent systems is one of the key research focuses. A prevalent learning framework is centralized training with decentralized execution (CTDE), in which the decentralized execution realizes strategy flexibility, and the use of centralized training ensures stationarity and goal consistency while becoming incapable when facing scalability and complicated situations. To address this issue, we follow the concept of distributed training with decentralized execution (DTDE). Decentralization is naturally accompanied by the game during the learning process, which has not been entirely studied in related work, resulting in the constrained strategy combination of MARL. In this paper, we devise a novel approach of differential reward interaction (DRI) with conflict-triggered for the distributed evaluation that enables overall goal consistency through highly efficient local information exchange. With this collaboration of learning, the DRI-based MARL can eliminate the notorious issue of converging to saddle equilibriums of stochastic games. Meanwhile, it possesses provable convergence and is well compatible for general value-based and policy-based algorithms. Experiments in several benchmark scenarios demonstrate that DRIMA realizes collaborative strategy learning with enhanced global goal-achieving.

## 1 INTRODUCTION

Multi-agent reinforcement learning (MARL) has been extensively studied in recent years owing to its capability of assisting multi-agent systems (MASs) in complex real-world applications, including multi-robot systems (Mou et al., 2021; Marchesini & Farinelli, 2022), autonomous driving (Palanisamy, 2020; Vinitsky et al., 2022), traffic signal control (Ault & Sharon, 2021; Du et al., 2023), and energy network regulation (Wang et al., 2021; Ma et al., 2024). MARL has been facing critical scalability issues since the size of the state and action space grows exponentially with the number of agents. The CTDE framework consists of centralized evaluation to collect information, such as observations, actions, or rewards of all agents, and decentralized execution that allows each agent to determine its own behavior according to individual observation. It largely mitigates the curse of dimensionality on the action space and has been adopted by many popular algorithms such as MADDPG (Lowe et al., 2017), VDN (Sunehag et al., 2017), COMA (Foerster et al., 2018), MAPPO (Yu et al., 2022), etc. Such a centralized evaluator, however, still has limitations in the scalability that assembles information from a mass of agents during training, and in complex collaborative tasks where the overall performance evaluator is insufficient to distinguish the global and local situation. A class of DTDE research on the distributed communication-based MARL then emerged, which mainly focuses on achieving global-level objectives with local-level collaborative learning.

Distributed MARL mainly differs in the way of information transmission and fusion. Qu and Lin (Qu et al., 2020; Lin et al., 2021; Qu et al., 2022) inventively proposed the exponential decay condition for general networked MARL that quantitatively bound the interaction strength between agents, under which the evaluation for each agent is scaled by the neighboring state-action space size instead of the global one, which realizes scalability. Mean Field Reinforcement Learning (MFRL) (Yang et al., 2018; Qiu et al., 2022; Yu, 2023) takes individual and average neighbor actions as

components of the respective state-action value function. The Mean-field theory, that each agent in a gridded world is influenced only by the mean effect of its neighbors, ensures respective value function is a good approximation to the joint one. On the other hand, the consensus-based method (Wai et al., 2018; Zhang et al., 2018; Qu et al., 2019; Chen et al., 2022) has been a popular topic in distributed MARL with parameterized functions. Zhang et al. (2018) pioneeringly introduces parameter consensus to the process of multi-agent learning. Based on the theory of distributed optimization and stochastic approximation, the achievement of global consistency by distributed value function has been proved. The convergence of this consensus approach is guaranteed for linear function approximation while is unreliable for neural networks (i.e. the nonlinear parameterized functions). There is also a class of works (Jiang et al., 2018; Blumenkamp & Prorok, 2021; Nayak et al., 2023) based on Graph Neural Networks (GNNs), designing sophisticated network structures to model multi-agent interactions and achieve collaboration in automatic learning ways.

An inevitable issue facing the decentralized decision-making MASs is the game between agents, which has not been fully studied in previous DTDE works and potentially leads the strategy combination to poor situations. Simultaneous independent decision-making and conflicting payoffs are two essential causes we identified. The former usually ensures strategy diversity, thus a promising direction is to work on the reconstruction of payoffs (or rewards) to solve the dilemma. Chu et al. (2020) proposed a reward reshaping method naturally based on the physical distance between neighboring agents for the network systems. Hostallero et al. (2020) designed a peer evaluation signal calculated by respective temporal difference (TD) that reflects the value of joint action, and agents receive it from peers to reshape their rewards. This method possesses more generality while less convergence property. Other learn-to-reshape methods were based on deep NNs (Yang et al., 2020; Yi et al., 2022), with special structures designed to automatically learn the reward-reshaping terms. The existing reshaping methods alleviate game dilemmas and promote cooperations to a certain extent, while the absence of identification and effective approaches to the critical causes results in limited generality and capability.

In this paper, a novel approach named DRIMA is proposed for distributed MARL that solves the dilemma of games for DTDE and achieves enhanced strategy collaborations. A conflict-triggered differential reward interaction (DRI) method is designed that allow agents to reshape personal rewards according to the opposite signal from neighbors. The conflict sign of DRs essentially reveals strategies contradiction, and the collaboration when it occurs enables the policy combinations to avoid saddle equilibriums in stochastic games. The interaction of DRs, which are scalars, compared with the transmission of NN parameters and the training of extra NN structures, realizes cooperative learning in a highly efficient way. DRIMA is naturally compatible with general multi-agent (deep) reinforcement learning algorithms and possesses provable convergence. The performance of our approach was examined on several benchmark scenarios, including the matrix games, Multi-Agent Particle Environment (MPE) (Lowe et al., 2017), and Star-Craft II (Samvelyan et al., 2019).

## 2 PRELIMINARIES

### 2.1 MULTI-AGENT MARKOV GAME

Multiple agents making sequential decisions in the same environment can be modeled as a $N$-agent Markov game (or $N$-player stochastic game) $\Gamma$, defined as a tuple $\langle \mathcal{S}, \mathcal{A}^1, \cdots, \mathcal{A}^N, r^1, \cdots, r^N, P \rangle$. $\mathcal{S}$ is the state space, $\mathcal{A}^i$ is the action space of agent $i$ $(i = 1, \cdots, N)$, $r^i : \mathcal{S} \times \mathcal{A}^1 \times \cdots \times \mathcal{A}^N \to \mathbb{R}$ is the reward function for agent $i$, and $P : \mathcal{S} \times \mathcal{A}^1 \times \cdots \times \mathcal{A}^N \to \Delta(\mathcal{S})$ is the transition probability map where $\Delta(\mathcal{S})$ is the set of probability distributions over state space $\mathcal{S}$. Agent policy is a mapping $\pi^i : \mathcal{S} \times \mathcal{A}^i \to \Delta(\mathcal{A}^i)$, where $\Delta(\mathcal{A}^i)$ denotes the space of probability distributions over the agent $i$'s actions. The joint policy of all agents is denoted as $\boldsymbol{\pi} = (\pi^1, \cdots, \pi^N)$, and the probability value of it is $\pi(\boldsymbol{a}|s) = \prod_{i=1}^N \pi^i(a^i|s)$. At each time step $t$, given current state $s_t$, each agent $i$ choose action $a_t^i$ according to own policy $\pi^i(\cdot|s_t)$. All agents form a joint action $\boldsymbol{a}_t = (a_t^1, \cdots, a_t^N)$ and each agent receives a reward $r^i(s_t, \boldsymbol{a}_t)$. The state then transits to the next $s_{t+1}$ based on the transition probability $P$ of the environment. Multiple agents in a Markov game will rationally try to find strategies that maximize their own expected returns. This paper aims to design a collaborative approach for distributed MARL to maximize the global return of MASs.

## 2.2 NETWORKED MARL FORMULATION

Networking is a common framework for studying MASs in their entirety. Agents in a networked system are denoted as a set $\mathcal{N} = \{1, \cdots, N\}$. The system can be modeled as an undirected graph $\mathcal{G}(\mathcal{N}, \mathcal{E})$, where each agent $i$ serves as vertex $i$ and $\mathcal{E} \subseteq \mathcal{N} \times \mathcal{N}$ is the set of all edges. The edge $e_{ij} = (i, j) \in \mathcal{E}$ indicates the connection relationship of two agents $i, j \in \mathcal{N}$. Two agents associated with an edge are neighbors, agent $i$ and its all neighbors in the graph together is denoted as a set $\mathcal{N}^i$. Note that the connection between agents could be explicit physical connection, communication relationship, implicit mutual influence, etc.

The global objective of the MAS is defined as maximizing the overall average reward per time step:

$$\underset{\pi}{\text{maximize}} \ J = \lim_T \frac{1}{T} \underset{s \sim P, \boldsymbol{a} \sim \boldsymbol{\pi}}{\mathbb{E}} \left( \sum_{t=0}^{T-1} \frac{1}{N} \sum_{i \in \mathcal{N}} r^i(s, \boldsymbol{a}) \right) = \sum_{s \in \mathcal{S}} d_{\boldsymbol{\pi}}(s) \sum_{\boldsymbol{a} \in \mathcal{A}} \pi(\boldsymbol{a}|s) \cdot \bar{r}(s, \boldsymbol{a}), \quad (1)$$

where $\bar{r}(\cdot) = \frac{1}{N} \sum_{i \in \mathcal{N}} r^i(s, \boldsymbol{a})$ is the average function of all agents' rewards.

To solve this optimization problem in a decentralized way, agents are set to make decisions independently. Based on the continuing RL problem formulation that utilizes the *average reward* in Sutton & Barto (2018), and considering that the action value of each agent $i$ reflects the expected value of its policy $\pi^i$ affected by other policies, the differential action-value function is formulated as:

$$Q^i_{\boldsymbol{\pi}}(s, a^i) = \sum_{t=0}^{\infty} \underset{s \sim P, \boldsymbol{a}^{-i} \sim \boldsymbol{\pi}^{-i}}{\mathbb{E}} \left[ r^i(s_t, \boldsymbol{a_t}) - \mu^i_{\boldsymbol{\pi}} | s_0 = s, a_0 = a^i \right],$$

where $\mu^i_{\boldsymbol{\pi}} = \lim_{T \to \infty} \frac{1}{T} \sum_{t=0}^{T-1} \mathbb{E}_{s \sim P, \boldsymbol{a} \sim \boldsymbol{\pi}} \left[ r^i(s_t, \boldsymbol{a_t}) \right] = \sum_{s \in \mathcal{S}} d_{\boldsymbol{\pi}}(s) \sum_{\boldsymbol{a} \in \mathcal{A}} \pi(\boldsymbol{a}|s) \cdot r^i(s, \boldsymbol{a})$ denotes the average reward per time step. $r^i(s_t, \boldsymbol{a_t}) - \mu^i_{\boldsymbol{\pi}}$ denotes the *differential reward* term of agent $i$. $d_{\boldsymbol{\pi}}(s)$ represents a stationary distribution over $\mathcal{S}$ induced by joint policy $\boldsymbol{\pi}$, which is assumed to exist in general problem formulations to ensure convergence theoretically (Zhang et al., 2018; Qu et al., 2020). $-i$ denotes agents excluding $i$. According to Bellman equation, the differential state-value function of $i$ is

$$v^i_{\boldsymbol{\pi}}(s) = \sum_{a^i \in \mathcal{A}^i} \pi^i(s, a^i) \cdot Q^i_{\boldsymbol{\pi}}(s, a^i).$$

Note that the average reward formulation can be generalized to the discount-reward one, that the differential reward term in above definitions is discounted by a time-decaying factor and becomes $\gamma^t \cdot (r^i(s_{t+1}, \boldsymbol{a}_{t+1}) - \mu^i_{\boldsymbol{\pi}}), \gamma \in [0, 1)$. In such setting, a discounted stationary distribution is assumed to exist (Nachum et al., 2019).

This paper focuses on general-sum games, which allows the agents' rewards to be arbitrarily related. Zero-sum games and coordination games are special cases where agents' rewards are always negatively and positively related. The baseline solution concept for general-sum games is Nash Equilibrium (NE), which denotes a joint strategy where each agent's response is the best to the others'. We focus on the special types of NE points: global optima, and saddles, which were well-studied in Hu & Wellman (2003). The detailed definitions refer to Appendix A.1. In the following, a collaborative evaluation method is proposed to avoid saddle equilibriums in Markov games for multi-agent reinforcement learning.

## 3 DIFFERENTIAL REWARD INTERACTION-BASED MARL

We identify that the global performance of distributed MARL is critically restricted by two factors. One is the simultaneous and independent decision-making, which ensures the diversity of strategy and the flexibility of decision, should be practically maintained. The other is the distinct payoffs individuals receive after joint action executions, which lead agents to make selfish evaluations and decisions that might be detrimental to the overall. We propose a method of Differential Reward Interaction (DRI) with conflict-triggered, whose key idea is based on the exchange of the average reward, a scalar reflecting a policy's overall performance, to identify and reshape conflicting payoffs between agents, that efficiently trade off joint goal consistency and individual diversity during learning and achieve the enhanced strategy combination.

### 3.1 Conflict-Triggered Differential Reward Interaction Method

For every agent $i$ $(i=1,\cdots,N)$ with original reward $r^i(s,\boldsymbol{a})$, abbreviated as $r^i$ in the following, the conflict-triggered interaction rule for differential rewards is designed as follows

$$\begin{cases} r^i - \bar{\mu}^{N^i}, \mathrm{sgn}\left(r^i - \bar{\mu}^{N^i}\right) = \mathrm{sgn}\left(\bar{r}^{N^{-i}} - \bar{\mu}^{N^i}\right) \\ \bar{r}^{N^i} - \bar{\mu}^{N^i}, \mathrm{sgn}\left(r^i - \bar{\mu}^{N^i}\right) \neq \mathrm{sgn}\left(\bar{r}^{N^{-i}} - \bar{\mu}^{N^i}\right) \end{cases}, \tag{2}$$

where $\mathrm{sgn}\left(\cdot\right)$ denotes Signum function

$$\mathrm{sgn}\left(x\right) = \begin{cases} -1 : x < 0 \\ 0 : x = 0 \\ 1 : x > 0 \end{cases},$$

$\bar{r}^{N^i}$ and $\bar{r}^{N^{-i}}$ are the averaged rewards of $i$'s neighbors with and without $i$ itself, defined as

$$\bar{r}^{N^i} = \frac{1}{N^i} \sum_{j \in \mathcal{N}^i} r^j \text{ and } \bar{r}^{N^{-i}} = \frac{1}{N^i - 1} \sum_{j \in \mathcal{N}^{-i}} r^j.$$

$\bar{\mu}^{N^i}$ indicates averaging the average reward of $i$'s neighbors, defined as

$$\bar{\mu}^{N^i} = \frac{1}{N^i} \sum_{j \in \mathcal{N}^i} \mu_{\boldsymbol{\pi}}^j,$$

where $\mu_{\boldsymbol{\pi}}^j$ is the time-average reward of agent $j$, i.e. , $\mu_{\boldsymbol{\pi}}^j = \lim_{T \to \infty} \frac{1}{T} \sum_{t=0}^{T-1} \mathbb{E}_{s \sim P, \boldsymbol{a} \sim \boldsymbol{\pi}}\left[r^j\left(s_t, \boldsymbol{a}_t\right)\right]$, $j \in \mathcal{N}^i$, and it can be incrementally estimated by respective reward samples through $\mu_t^j \leftarrow \mu_{t-1}^j + \alpha_t \cdot \left(r^j\left(s_t, \boldsymbol{a}_t\right) - \mu_{t-1}^j\right)$ where $\alpha_t$ is an appropriate step-size.

In Eqn. (2), the opposite sign of the differential reward terms in the Signum function indicates the "conflict" between agents about their value assessments of the group's behaviors. To get an intuition of this idea, consider first a ***single-agent*** system. At each iteration step $t$ $(t=1,2,\cdots)$, the agent obtains a differential reward $r\left(s_t, a_t\right) - \mu_t$ according to the instant reward sample. Since $\mu_t$ estimates the expected average reward and reflects the overall assessment of the current policy, the immediate differential reward in fact indicates a (positive or negative) contribution value to the current policy by taking action $a_t$ at state $s_t$. Relating to the ***multi-agent*** system, based on Eqn. (2), each agent $i$ is allowed to constantly communicate average rewards with neighbors to obtain $\bar{\mu}_t^{N^i}$, which reflects the performance on average of current neighbor-joint policy $\boldsymbol{\pi}_t^{N^i}$. And the opposite sign between $r^i\left(s_t, \boldsymbol{a}_t\right) - \bar{\mu}_t^{N^i}$ and $\bar{r}^{N^{-i}}\left(s_t, \boldsymbol{a}_t\right) - \bar{\mu}_t^{N^i}$ indicates the conflict contribution of action decision relating to the individual $i$ and its other neighbors $N^{-i}$ to the current neighbor-joint policy. When conflict signs occur, the agent's personal reward $r^i\left(s_t, \boldsymbol{a}_t\right)$ be reshaped as the neighbor-averaged reward $\bar{r}^{N^i}\left(s_t, \boldsymbol{a}_t\right)$ to keep joint goal consistency. Based on differential reward interaction and reward consensus at critical steps, the conflict-triggered DRI method enables agents to maintain collaboration during learning, balance joint goal consistency and individual behavior diversity, and eliminate joint strategy to saddle equilibriums in a highly efficient way. We present the detailed study in the following.

### 3.2 Two-Player Matrix Game

Without loss of generality, a Prisoner's dilemma game is provided with payoffs (i.e. rewards) in Fig. 1. It originally has a unique NE which is a saddle point valued at $(2, 2)$, and the global optimum, valued at $(10, 10)$, is not an NE. For the problem formulation of MARL, this game can be seen as a continuing task that is repeatedly played by two agents. It is a non-state transition task, i.e., it can be regarded as having only one state $s_0$, and no matter what joint action performed at each step will lead to the same state. The action spaces of Red and Blue agents are $\mathcal{A}^{\mathrm{re}}, \mathcal{A}^{\mathrm{bl}} = \{a_C, a_D\}$, where $a_C$ means cooperate and $a_D$ is defect. The joint action is denoted as $\boldsymbol{a} = \left(a^{\mathrm{re}}, a^{\mathrm{bl}}\right), \boldsymbol{a} \in \mathcal{A}$ where $\mathcal{A} = \mathcal{A}^{\mathrm{re}} \times \mathcal{A}^{\mathrm{bl}}$. Following the networked formulation, two agents are set to be neighbors and allow

| Red \\ Blue | Cooperate | Defect |
|---|---|---|
| Cooperate | 10 \\ 10 | 10 \\ 18 |
| Defect | 18 \\ 0 | 0 \\ 2 |

Figure 1: Prisoner's dilemma payoff matrix.

communication. According to DRI method, their action and state value functions are as

$$\tilde{Q}_{\boldsymbol{\pi}}^i\left(s_0, a^i\right) = \sum_{t=0}^{\infty} \mathop{\mathbb{E}}_{a^{-i}\sim\pi^{-i}}\left[\tilde{r}^i(s_0, \boldsymbol{a}_t) - \bar{\mu}_{\boldsymbol{\pi}}|s_0, a_0 = a^i\right] \text{ and } \tilde{v}_{\boldsymbol{\pi}}^i\left(s_0\right) = \sum_{a^i\in\mathcal{A}^i}\pi^i\left(a^i|s_0\right)\cdot\tilde{Q}_{\boldsymbol{\pi}}^i\left(s_0, a^i\right),$$

where $\tilde{r}_t^i, i \in \{\text{re, bl}\}$ is the reshaped reward following the interaction rule Eqn. (2), and $\bar{\mu}_{\boldsymbol{\pi}} = \left(\mu_{\boldsymbol{\pi}}^{\text{re}} + \mu_{\boldsymbol{\pi}}^{\text{bl}}\right)/2$. The objective of this two-agent system is

$$\operatorname*{maximize}_{\boldsymbol{\pi}_*} J_* = \sum_{\boldsymbol{a}\in\mathcal{A}} \pi_*(\boldsymbol{a})\cdot\frac{1}{2}\left(r^{\text{re}}(\boldsymbol{a}) + r^{\text{bl}}(\boldsymbol{a})\right) = \frac{1}{2}\left(\mu_{\boldsymbol{\pi}_*}^{\text{re}} + \mu_{\boldsymbol{\pi}_*}^{\text{bl}}\right) = \bar{\mu}_{\boldsymbol{\pi}_*},$$

where $\boldsymbol{\pi}_* = \left(\pi_*^{\text{re}}, \pi_*^{\text{bl}}\right)$ is Nash equilibrium.

We take the policy gradient Actor-Critic algorithm here to analyze the learning process to the equilibrium. Set two decision-making policies defined by two parameters $p$ and $q$, respectively, as

$$\begin{cases}\pi^{\text{re}}(a_C) = 1-p \\ \pi^{\text{re}}(a_D) = p\end{cases}, \quad \begin{cases}\pi^{\text{bl}}(a_C) = 1-q \\ \pi^{\text{bl}}(a_D) = q\end{cases}, \quad p, q \in (0, 1). \tag{3}$$

$\boldsymbol{\pi} = \left(\pi^{\text{re}}, \pi^{\text{bl}}\right)$ denotes the joint strategy. For each agent $i$, Actor-Critic iteratively advances by a faster Critic step to estimate value function $\tilde{v}_{\boldsymbol{\pi}_t}^i$, alternating with a slower Actor step for policy $\pi_t^i$ improvement based on current value function. At the $t$-th iteration, regarding the faster speed of the evaluation step, the policy improvement step can be analyzed separately under the assumption that $\tilde{v}_{\boldsymbol{\pi}_t}^i$ and $\bar{\mu}_{\boldsymbol{\pi}_t}$ of the current policy $\boldsymbol{\pi}_t$ are already the expected values. To the value function defined by the differential return, $\mathbb{E}_{s\in\mathcal{S}}[v_\pi(s)] = 0$ holds (Sutton & Barto, 2018). We have $\tilde{v}_{\boldsymbol{\pi}_t}^i(s_0) = 0$ in this case, thus the one-step TD term $\tilde{r}^i(\boldsymbol{a}_t) - \bar{\mu}_{\boldsymbol{\pi}_t} + \tilde{v}_{\boldsymbol{\pi}_t}^i(s_{t+1}) - \tilde{v}_{\boldsymbol{\pi}_t}^i(s_t)$ degenerates as $\tilde{r}^i(\boldsymbol{a}_t) - \bar{\mu}_{\boldsymbol{\pi}_t}$. The policy parameter update for each agent at $t$ is then as follows

$$\begin{aligned} p_{t+1} &= p_t + \beta_t\cdot(\tilde{r}^{\text{re}}(\boldsymbol{a}_t) - \bar{\mu}_{\boldsymbol{\pi}_t})\cdot\nabla_p\log\pi_{p_t}^{\text{re}}(a_t^{\text{re}}), \\ q_{t+1} &= q_t + \beta_t\cdot\left(\tilde{r}^{\text{bl}}(\boldsymbol{a}_t) - \bar{\mu}_{\boldsymbol{\pi}_t}\right)\cdot\nabla_q\log\pi_{q_t}^{\text{bl}}(a_t^{\text{bl}}), \end{aligned} \tag{4}$$

where $\beta_t$ is an appropriate step-size of the Actor. Based on the policy definition of Eqn. (3), and to facilitate analysis, the gradient terms in Eqn. (4) can always be transferred to be positive no matter what action is sampled. With this view, the differential reward terms are directly affects the gradient ascent direction and size. We draw the geometry of this optimization problem in Fig. 2 to illustrate the conflict-triggered mechanism of DRI.

As shown in Fig. 2(a), the x-axis and y-axis represent the domains of $p$ and $q$, respectively, and the z-axis represents the value range of $\mu_{\boldsymbol{\pi}}$. In feasible region, $\mu_{\boldsymbol{\pi}}^{\text{re}}$ is displayed as the Hot-colored surface, and $\mu_{\boldsymbol{\pi}}^{\text{bl}}$ is as Cold-colored surface. $\bar{\mu}_{\boldsymbol{\pi}}$ denotes the average of $\mu_{\boldsymbol{\pi}}^{\text{re}}$ and $\mu_{\boldsymbol{\pi}}^{\text{bl}}$ shown as the Rainbow-colored one between them. One can see that independent learning will lead the Red and Blue agent approach to $p = 1$ and $q = 1$ (valued at $(2, 2)$) along the x-axis and y-axis respectively, since it is the highest point in the direction of their respective gradient ascent. But the global optimum, valued at $(10, 10)$ when $p = 0, q = 0$, is the highest point of the joint policy solution on Rainbow surface $\bar{\mu}_{\boldsymbol{\pi}}$.

Conflict-triggered differential reward interaction allows each agent to reconstruct the joint surface at the necessary steps and approach the joint optimum. To see this, we arbitrarily assume that the current policy is at $p_t = 0.4, q_t = 0.8$, then $\bar{\mu}_{\boldsymbol{\pi}_t} = 6.88$ is depicted as a green triangle on the Rainbow surface. Agents take actions with the probabilities based on respective current policies. And the

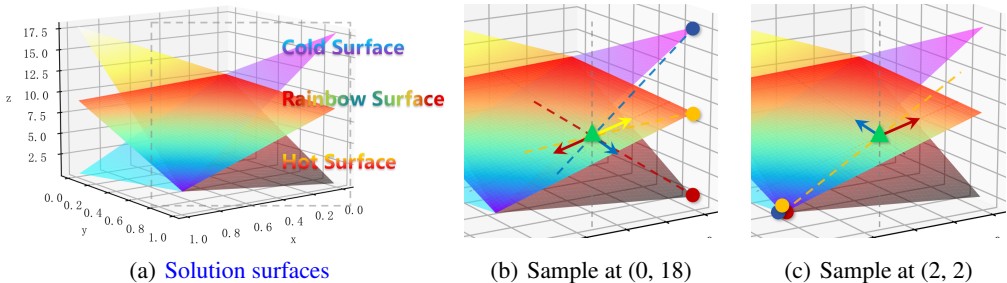

(a) Solution surfaces      (b) Sample at (0, 18)      (c) Sample at (2, 2)

Figure 2: Solution space visualization of the Prisoner's Dilemma.

formed joint action $\boldsymbol{a}_t = \left(a_t^{\mathrm{re}}, a_t^{\mathrm{bl}}\right)$ produces a related joint reward sample $\left(r^{\mathrm{re}}\left(\boldsymbol{a}_t\right), r^{\mathrm{bl}}\left(\boldsymbol{a}_t\right)\right)$, i.e. one of (10, 10), (0, 18), (18, 0), and (2, 2) according to the payoff matrix. The conflict-triggered mechanism works when the two differential reward values calculated by the associated reward samples possess different signs, i.e. $\mathrm{sign}\left(r_t^{\mathrm{re}} - \bar{\mu}_{\boldsymbol{\pi}_t}\right) \neq \mathrm{sign}\left(r_t^{\mathrm{bl}} - \bar{\mu}_{\boldsymbol{\pi}_t}\right)$. Here, samples (0, 18) and (18, 0) trigger interactions, while samples (10, 10) and (2, 2) do not.

Taking (0, 18) for an example, it and its average value of 9 are shown in red, blue, and yellow dots in Fig. 2(b). The corresponding differential values for 0, 18, 9 are -6.88, 11.12, 2.12, that indicates the sample's contribution to joint policy, from the view of the Red, Blue, and (hypothetic) global agent. Three colored dashed lines represent the respective sight line of each dot observing the current $\bar{\mu}_{\boldsymbol{\pi}_t}$ (drawn as the green triangle). One can see that, on the Rainbow joint surface, from the global view (i.e., the yellow dot), the gradient ascent direction projection along the x-axis and y-axis (that respectively indicate the policy parameters' feasible direction for agent Red and Blue) are the yellow and blue vectors. However, the ascent direction from the Red dot view on the Hot surface, which is along the red dash line, projects to its feasible direction and then to the joint surface (drawn as the red vector), opposite to the global view (the yellow vector). That is to say, the direction of Red agent's strategy optimization, based on the current personal differential reward sampling, is contrary to the global optimization direction.

Specifically, conflict arises due to opposite viewing angles when two points above and below the horizontal of the $\bar{\mu}_{\boldsymbol{\pi}_t}$-point observe it respectively, causing goal inconsistency. The opposite sign of differential rewards between agents exactly represents such conflict. When it occurs, agents exchange and average respective differential rewards through the method of Eqn. (2) to obtain a global view, thus maintaining a consistent perspective of the global goal during learning. Moreover, agents still use respective original rewards to learn for those samples that do not cause conflicts, such as (2, 2) (shown in Fig. 2(c)), more generally, its value could be $(a, b), 0 < a, b < 9$ in this game. This is critical for the generalization of tasks with dense rewards, and the achievement of efficient trade-offs between local-level and global-level learning. Independent learning, at non-conflict steps, enables agents to fully explore respective strategies that promise robust strategy combinations with enhanced performances. The contrastive ablation experiments conducted in MPE (Section 4.2) validated this.

**Remark 1.** It can be inferred that, for general multi-dimensional action spaces that are (linearly or nonlinearly) denoted with multi-dimensional parameterized policies, the conflict-triggered mechanism still works in high-dimensional spaces.

### 3.3 MULTI-PLAYER MARKOV GAME

Section 3.2 investigates the problem formulated as a repeated matrix game involving two players, without considering state transitions. For general Markov games with a number of participants, in which state transitions are involved and rewards are associated with state-action pairs at each step, we next show that DRI method remains valid. The mean-field theory (MFT) (Stanley, 1971) is employed here to analyze agents in a networked system, it claims that each agent is affected only by the mean effect from others around it that converts many-agent interactions into two-agent interactions. Since multi-agent MDP evolves in a networked environment, the pairwise approximation still preserves global interactions between any pair of agents implicitly (Blume, 1993; Yang et al., 2018). Note that Yang et al. (2018) utilize MFT to construct respective scalable joint action-value functions

by assembling the average neighbor actions to achieve collaborative evaluation. This method is still incapable of dealing with saddle equilibriums due to value functions are learned by distinct payoffs. We further exploit the MFT concept to reconstruct payoffs for value functions. Accordingly, the conflict detection and reward reshaping of DRI method work between each agent $i, i \in \mathcal{N}$ and its averaged others in $\mathcal{N}^{-i}$, where $\mathcal{N}^{-i}$ denoted the set of agents except $i$. Let $j$ denotes the averaged $\mathcal{N}^{-i}$, and consider two agents $i$ and $j$ with the state value functions as

$$v_{\boldsymbol{\pi}}^i(s) = \sum_{t=0}^{\infty} \mathbb{E}_{s \sim P, \boldsymbol{a} \sim \boldsymbol{\pi}} \left[ \left( r^i(s_t, \boldsymbol{a}_t) - \bar{\mu}_{\boldsymbol{\pi}} | s_0 = s \right) \right] \text{ and } v_{\boldsymbol{\pi}}^j(s) = \sum_{t=0}^{\infty} \mathbb{E}_{s \sim P, \boldsymbol{a} \sim \boldsymbol{\pi}} \left[ \left( r^j(s_t, \boldsymbol{a}_t) - \bar{\mu}_{\boldsymbol{\pi}} | s_0 = s \right) \right],$$

where $\bar{\mu}_{\boldsymbol{\pi}} = \frac{1}{2} \left( \mu_{\boldsymbol{\pi}}^i + \mu_{\boldsymbol{\pi}}^j \right)$.

In cases, the previous matrix game for example, where the reward scale of both agents is equal, i.e., $\frac{1}{|\mathcal{S}| \cdot |\mathcal{A}|} \sum_{s \in \mathcal{S}, \boldsymbol{a} \in \mathcal{A}} r^i(s, \boldsymbol{a}) = \frac{1}{|\mathcal{S}| \cdot |\mathcal{A}|} \sum_{s \in \mathcal{S}, \boldsymbol{a} \in \mathcal{A}} r^j(s, \boldsymbol{a})$, we have $\mathbb{E} \left[ v_{\boldsymbol{\pi}}^i(s) \right] = \mathbb{E} \left[ v_{\boldsymbol{\pi}}^j(s) \right] = 0$. In general cases, we have $\mathbb{E} \left[ v_{\boldsymbol{\pi}}^i(s) \right] = \frac{\mu_{\boldsymbol{\pi}}^i - \mu_{\boldsymbol{\pi}}^j}{2}, \mathbb{E} \left[ v_{\boldsymbol{\pi}}^j(s) \right] = \frac{\mu_{\boldsymbol{\pi}}^j - \mu_{\boldsymbol{\pi}}^i}{2}$. To our method, the state value $\tilde{v}$ is defined by the reshaped reward $\tilde{r}$ of Eqn. (2), then we can obtain

$$\begin{cases} \bar{\mu}_{\boldsymbol{\pi}} = \frac{\tilde{\mu}_{\boldsymbol{\pi}}^i + \tilde{\mu}_{\boldsymbol{\pi}}^j}{2}, & \mathbb{E} \left[ \tilde{v}_{\boldsymbol{\pi}}^i(s) \right] = \frac{\tilde{\mu}_{\boldsymbol{\pi}}^i - \tilde{\mu}_{\boldsymbol{\pi}}^j}{2}, & \mathbb{E} \left[ \tilde{v}_{\boldsymbol{\pi}}^j(s) \right] = \frac{\tilde{\mu}_{\boldsymbol{\pi}}^j - \tilde{\mu}_{\boldsymbol{\pi}}^i}{2}, \\ \mathbb{E} \left[ \tilde{v}_{\boldsymbol{\pi}_t}^i(s_{t+1}) - \tilde{v}_{\boldsymbol{\pi}_t}^i(s_t) \right] = \mathbb{E} \left[ \tilde{r}^i(\boldsymbol{a}_t) \right] = \tilde{\mu}_{\boldsymbol{\pi}}^i, & \mathbb{E} \left[ \tilde{v}_{\boldsymbol{\pi}_t}^j(s_{t+1}) - \tilde{v}_{\boldsymbol{\pi}_t}^j(s_t) \right] = \mathbb{E} \left[ \tilde{r}^j(\boldsymbol{a}_t) \right] = \tilde{\mu}_{\boldsymbol{\pi}}^j, \end{cases}$$

where $\tilde{\mu}_{\boldsymbol{\pi}}^i, \tilde{\mu}_{\boldsymbol{\pi}}^j$ are the reshaped average reward. Now, the general one-step TD policy parameter update at step $t$ is as follows

$$\begin{aligned} \theta_{t+1}^i &= \theta_t^i + \beta_t \cdot \left[ \tilde{r}_t^i - \bar{\mu}_{\boldsymbol{\pi}_t} + \tilde{v}_{\boldsymbol{\pi}_t}^i(s_{t+1}) - \tilde{v}_{\boldsymbol{\pi}_t}^i(s_t) \right] \cdot \nabla_{\theta_t^i}, \\ \theta_{t+1}^j &= \theta_t^j + \beta_t \cdot \left[ \tilde{r}_t^j - \bar{\mu}_{\boldsymbol{\pi}_t} + \tilde{v}_{\boldsymbol{\pi}_t}^j(s_{t+1}) - \tilde{v}_{\boldsymbol{\pi}_t}^j(s_t) \right] \cdot \nabla_{\theta_t^j}, \end{aligned} \tag{5}$$

where $\nabla_{\theta_t^{\cdot}} = \nabla_{\theta_t^{\cdot}} \log \pi(a_t^{\cdot} | s_t)$ denotes the respective policy gradient term. One can see that in Eqn. (4), the conflict detection mechanism works on the horizontal $\bar{\mu}_{\boldsymbol{\pi}_t}$. While in Eqn. (5), it works on the horizontal $\bar{\mu}_{\boldsymbol{\pi}_t} - \tilde{\mu}_{\boldsymbol{\pi}_t}^i$ for agent $i$ and the $\bar{\mu}_{\boldsymbol{\pi}_t} - \tilde{\mu}_{\boldsymbol{\pi}_t}^j$ for agent $j$, it accurately eliminates the mismatch of different reward scales. An exponential decay concept was investigated by Qu et al. (2020) that ensures the scalability and provable convergence of networked MARL. Based on this work, we obtain the convergence Proposition of DRIMA, theoretical details are presented in Appendix A.2.

Note that the Nash equilibrium approached by DRIMA is the reshaped one of the original game. DRI method reshapes original payoff $(Q_t^1, \cdots, Q_t^N)$ to $(\tilde{Q}_t^1, \cdots, \tilde{Q}_t^N)$, $t = 1, 2, \cdots$, that reshapes the solution space of each agent to avoid the saddle equilibrium point. Hence, the stationary points of solution space only contains local optimum, global optimum, and inflection ones. Compared with the saddle points that decentralized executive MARL generally can not escape, the sub-optimal points can be avoided. The ability of sub-optimal avoidance mainly relies on the stochastic gradient algorithm agents utilized, the nonlinear level of solution space concerning the task complexity, and the way of collaboration. Conflict-triggered DRI enables the joint strategy to approach better optimum with effective collaboration, Appendix B.1 presents an experimental examination. The pseudo-code of DRI-based Actor-Critic is provided in Appendix C.

## 4 EXPERIMENTS

In this section, experiments of three scenarios are presented – the matrix game, the Multi-Agent Particle Environment (MPE) (Lowe et al., 2017) and the StarCraft Multi-Agent Challenge (SMAC), containing state and action spaces as discrete to continuous. DRIMA-based algorithms are labeled as '-Dri', CTDE labeled as '-Ctde', and independent ones as '-Ind'. To draw training curves, all algorithms run 5 times using 5 random seeds. All the value and policy functions in the matrix game and MPE are parameterized by Multi-Layer Perceptrons without the recurrent structures and leave off the parameter sharing to diminish interfering factors. For SMAC where the scenario and strategy complexity increased, the recurrent NN embeds historical observations and parameter sharing ensures training efficiency, we adopt these two techniques to test the efficacy of DRIMA. Experiments are built on the PARL framework[1] and MAPPO benchmark (Yu et al., 2022), and our code will be available on GitHub after this article is published.

---

[1]https://github.com/PaddlePaddle/PARL

### 4.1 MATRIX GAME

Table 1: Maintain game where B-Y and C-Z are the NE, A-X is the global optimum.

|   | X | Y | Z |
|---|---|---|---|
| A | 20, 15 | 0, 0 | 0, 0 |
| B | 30, 0 | 10, 5 | 0, 0 |
| C | 0, 0 | 0, 0 | 5, 10 |

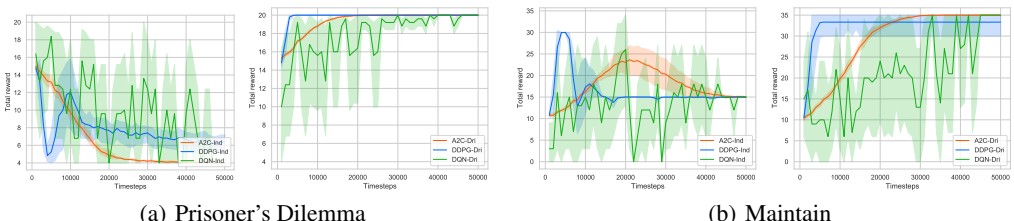

(a) Prisoner's Dilemma         (b) Maintain

Figure 3: Total rewards in matrix games.

Two 2-player matrix games, the Prisoner's Dilemma and the Maintain (Zhang et al., 2020), are tested here. Their payoffs are shown in Fig. 1 and Table 1 respectively. The Dilemma has a unique NE, in this case a saddle point, valued at (2, 2), while (10, 10) is the global optimum. For the Maintain in which each agent has 3 actions, two NEs are valued at (10, 5) and (5, 10). The global optimum at (20, 15) is a Pareto optimality but not a Nash equilibrium. Fig. 3(a) shows the learning curves of total reward related to the independent and DRIMA algorithms in Dilemma, and Fig. 3(b) denotes the results of Maintain. One can see that, the curves of value-based DQN show large fluctuations due to the lack of strategy exploration and stochasticity. There always is a certain probability of exploration, predefined by a (usually decreasing) factor $\epsilon$, to the points that make the deviation of trajectories. Overall, DRIMA enables DQN-Dri approaches to the global optimum in two tasks, while independent DQNs all converge to poor equilibriums. For the deterministic policy, DDPG-Ind converge to the saddle NE in both tasks. With the DRI method, DDPG-Dri converges to the global optimum in Dilemma. In Maintain, it converges to a probability distribution of optimal and suboptimal points due to insufficient exploration, but the result is still superior to saddle NE. The stochastic policy A2C performs best with DRI that achieves the global optimum, without it, A2C-Ind falls into saddle points. Note that in Zhang et al. (2020), the optimum is achieved through a centralized Bi-level training approach and requires the joint action vector. A 3-player matrix game was further conducted, see Appendix B.1 for more discussion.

### 4.2 MULTI-AGENT PARTICLE ENVIRONMENT

For continuous space scenarios, we examine DRIMA in MPE. In this case, the interactions are allowed in a certain communication range, and agents are practically observe state from their own perspective. Two types of tasks named Cooperative Hunting and Cooperative Collection are designed shown in Fig. 4(a) and (d) respectively. For Cooperative Hunting, the predators (blue and green) are slower and try to catch the faster prey (red). The prey's moving strategy is pre-trained on the rules to avoid being hit, it is allowed to observe within a limited view (marked by the red area) to ensure positive activity. Two types of agents consist of the predator group, in which the attacker (blue) is assumed to have the firepower to hit the prey and finish the task, while the interceptors (greens) are poorly equipped which can only hinder the movement of the target and avoid being hit. The predator group needs to learn cooperative strategies to hit prey within a limited episode length (30 steps). The rewards for all predators are set to negative values of their respective distances to the prey, with the attacker receiving an additional positive reward for hitting the prey. The saddle situation it might encounter is discussed in Appendix B.2.1 in detail.

We first design a hunting task, CH-I, composed of 1 prey, 1 attacker, and 2 interceptors. For multi-agent DDPG and A2C, in CH-I, the predator group reward over the number of episodes are shown

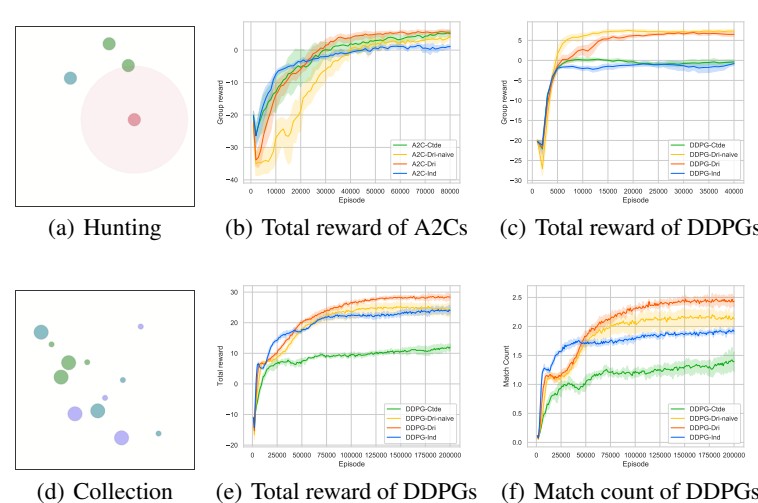

(a) Hunting     (b) Total reward of A2Cs     (c) Total reward of DDPGs

(d) Collection     (e) Total reward of DDPGs     (f) Match count of DDPGs

Figure 4: Experimental results in Multi-Agent Particle Environment.

as Fig. 4 (b) and (c). One can see that both A2C-Dri and DDPG-Dri successfully learned the group strategy of hitting the prey that obtains positive rewards. Based on A2C-Ind and DDPG-Ind, however, the predator group can hardly achieve the capture goal and the final reward is almost zero. The '-Ctde', despite using a global reward that sums up all agents' rewards for learning, cannot guarantee learning a globally optimal policy. A2C-Ctde succeeded and DDPG-Ctde failed in this task. The reason might be that higher stochasticity brings A2C moving policy greater flexibility and probability to explore a successful combination. The lack of stochasticity and reward distinction between multiple agents might cause the failure of DDPG-Ctde. We further design a CH-II composed of 2 prey, 2 attackers, and 2 interceptors to exam an increased scale. Considering limited space, we put the demonstration of the learned group policy, and the results of CH-II in Appendix B.2.2.

For Cooperative Collection (Fig. 4(d)), 6 agents are set to search and occupy 6 randomly located food. Both agents and food have three types, purple, blue, and green, and set the number of each type as 2. Agents need to collaboratively occupy as much food as possible and, meanwhile, occupy the food that matches their type as much as possible during one episode. Each agent can only observe objects within its field of view, it receives a negative reward based on the distance to the nearest food location, a positive reward when it occupies any food, and a maximum reward if the occupied food is of the same type. The saddle situation it might encounter is discussed in Appendix B.2.1 in detail.

Fig. 4(e) and (f) present the learning curves of DDPGs. The '-Dri', compared with '-Ctde' and '-Ind', achieves the best overall performance, and it does learn a collaborative strategy of occupying as much food as possible while promoting agent-food type matching (Fig. 4(f)). Interestingly, a class of A2C algorithms, which performs well in Cooperative Hunting, performs poorly in Cooperative Collection. We attribute this to the existence of symmetric games, in which the identities of the players can be changed without changing the payoff to the strategies. It occurs in tasks such as occupying the same target and collision avoidance between homogeneous agents. More specifically, if in a continuous space environment with multiple homogeneous agents, each adopts a policy with high stochasticity, it would be difficult for agents to break the symmetric situation encountered. We provide a detailed discussion in Appendix B.2.3.

Meanwhile, the '-Dri-naive' algorithms that constantly average neighbors' rewards without the conflict trigger are performed, in contrast to the '-Dri'. For the previous matrix games with simple reward constructions, the constant average method can also obtain results comparable with DRIMA. In the continuous space MPE, however, the general inferior performance of '-Dri-naive' shows the critical role of the conflict-triggered mechanism. The greater the scale and complexity of a task, the more significant it is (Fig. 4 and Fig. 7). Integrating information only when necessary can effectively balance individual-level and global-level learning, helping to learn a diversified and robust policy combination, that ensures an enhanced performance of MARL.

### 4.3 STARCRAFT MULTI-AGENT CHALLENGE

In this section, based on the MAPPO benchmark (Yu et al., 2022), we perform DRIMA combined with MAPPO in StarCraft II. In this battle case, agents in the Ally group strive to learn collaborative strategies to eliminate all agents in the Enemy group to win. All the learning agents are set to have access to a group reward that reflects the overall health status of the Enemy group. The distinctions are that for the '-Ctde', each agent receives a reward containing all ally members, while the '-Ind' and '-Dri' agents receive a reward only reflecting individual health status, moreover, the '-Dri' allows differential reward interactions within sight ranges.

Several representative tasks are performed, covering the homogeneous and symmetrical, homogeneous and asymmetrical, heterogeneous and symmetrical, and heterogeneous and asymmetrical formations from the perspective of role composition and number of members; crossing easy, hard, and super hard levels of map difficulty. The median win rate curves are shown in Fig. 5. It can be seen that in general tasks, the DRI algorithms possess a relatively slow rate at the beginning of learning. One can infer that distributed information interactions for goal consensus during the learning process would affect the learning rate within an acceptable range. The learning results, nevertheless, were not degraded. DRIMA based on its effective collaboration achieves at least comparable results with the centralized benchmark MAPPO-Ctde, in a distributed way. It should be noticed that in the hard task 5m vs. 6m, DRIMA achieves an outperforming win rate, and we attribute this to its saddle elimination capability. For complicated multi-agent tasks in which it is difficult to identify the saddle situations constructed by original reward functions, DRIMA enables MARL to avoid potential saddle equilibriums during learning and approach to better optimum. As to the Ctde, a global reward setting in such tasks may make it difficult to distinguish the individual contribution from the joint behavior and to promote strategy combination.

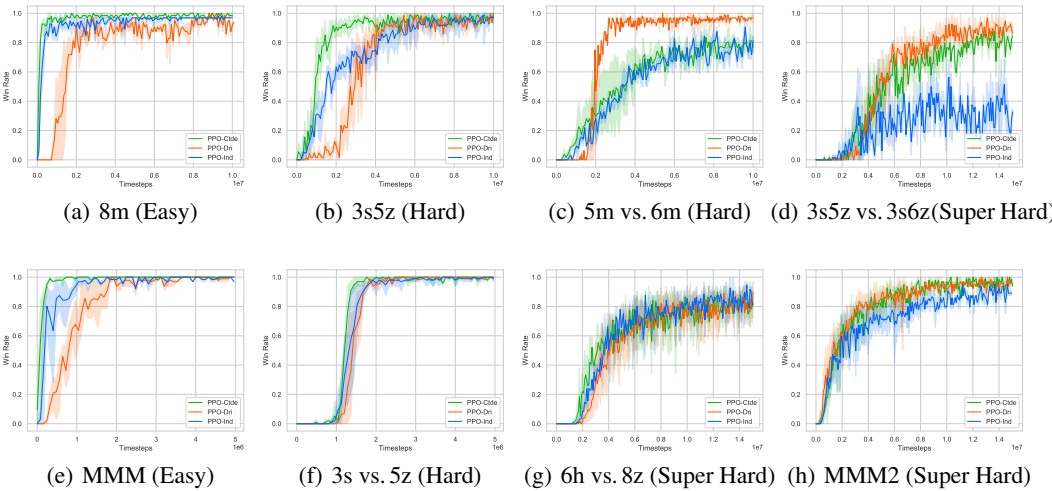

(a) 8m (Easy)  (b) 3s5z (Hard)  (c) 5m vs. 6m (Hard)  (d) 3s5z vs. 3s6z (Super Hard)

(e) MMM (Easy)  (f) 3s vs. 5z (Hard)  (g) 6h vs. 8z (Super Hard)  (h) MMM2 (Super Hard)

Figure 5: Results in SMAC.

## 5 CONCLUSION

In this paper, we introduced DRIMA, a cooperative learning approach based on the conflict-triggered differential reward interaction for distributed MARL. We identify that the game induced by decentralized executions of multiple agents tends to limit overall performance, our approach efficiently eliminates strategies from converging to saddle equilibriums to solve this dilemma. Distributed learning promises great flexibility and robust capability, achieving ideal collaboration performance on this basis is an ongoing challenge. In the future, we will continue to study the remaining issues, such as breaking the symmetry of the game for multi-agent stochastic policies.

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

# A  THEORETICAL MATERIALS

## A.1  DEFINITIONS OF NASH EQUILIBRIUMS

The Nash equilibrium of the stochastic game denotes a joint strategy where each agent's response is the best to the others', according to Hu & Wellman (2003), the definition is as follows.

**Definition 1.** (**Nash equilibrium**) In stochastic game $\Gamma$, a Nash equilibrium point is a tuple of $N$ strategies $\left(\pi_*^1, \cdots, \pi_*^N\right)$ such that for all $s \in \mathcal{S}$ and $i = 1, \cdots, N$,

$$v_{\pi_*^1, \cdots, \pi_*^N}^i(s) \geq v_{\pi_*^1, \cdots, \pi_*^{i-1}, \pi^i, \pi_*^{i+1}, \cdots, \pi_*^N}^i(s),$$

for all $\pi^i \in \Pi^i$, where $\Pi^i$ is the set of strategies available to agent $i$. Correspondingly, the Nash $Q$-function of agent $i$ is

$$Q_*^i\left(s, a^1, \cdots, a^N\right) = r^i\left(s, a^1, \cdots, a^N\right) + \sum_{s' \in S} P\left(s' | s, a^1, \cdots, a^N\right) v^i\left(s', \pi_*^1, \cdots, \pi_*^N\right),$$

where $r^i\left(s, a^1, \cdots, a^N\right)$ is agent $i$'s one-period reward in state $s$ and under joint action $\left(a^1, \cdots, a^N\right)$, $v^i\left(s', \pi_*^1, \cdots, \pi_*^N\right)$ is agent $i$'s expected reward over infinite periods starting from state $s'$ given that agents follow the equilibrium strategies.

The strategies are assumed to be stationary, that is, the rule of choosing an action is the same in every stage $t$, to ensure the existence of the Nash equilibrium point. Meanwhile, a stochastic game can be seen as composed of many stage games (one-period games) with the state transition. We focus on stage games with special types of Nash equilibrium points: global optima, and saddles, the related definitions are given.

**Definition 2.** (**Global optimal point**) A joint strategy $\left(\sigma^1, \cdots, \sigma^N\right)$ of the stage game $\left(M^1, \cdots, M^N\right)$ is a global optimal point if every agent receives its highest payoff at this point. That is, for all $k$,

$$\sigma M^k \geq \hat{\sigma} M^k \text{ for all } \hat{\sigma} \in \sigma(\mathcal{A}).$$

A global optimal point is always a Nash equilibrium.

**Definition 3.** (**Saddle point**) A joint strategy $\left(\sigma^1, ..., \sigma^N\right)$ of the stage game $\left(M^1, ..., M^N\right)$ is a saddle point if (1) it is a Nash equilibrium, and (2) each agent would receive a higher payoff when at least one of the other agents deviates. That is, for all $k$,

$$\sigma^k \sigma^{-k} M^k \geq \hat{\sigma}^k \sigma^{-k} M^k \text{ for all } \hat{\sigma}^k \in \sigma\left(\mathcal{A}^k\right),$$
$$\sigma^k \sigma^{-k} M^k \leq \sigma^k \hat{\sigma}^{-k} M^k \text{ for all } \hat{\sigma}^{-k} \in \sigma\left(\mathcal{A}^{-k}\right).$$

$M^k$ is agent $k$'s payoff function over the space of joint actions, $M^k = \left\{r^k\left(a^1, ..., a^N\right) | a^1 \in \mathcal{A}^1, ..., a^N \in \mathcal{A}^N\right\}$, and $r^k$ is the reward of agent $k$.

## A.2  CONVERGENCE ANALYSIS OF DRIMA

Qu et al. (2020) investigated the exponential decay property that ensures the scalability of networked MARL through the proposed theoretical conditions. Following this work, we obtain the Lemma below.

**Lemma 1.** *Define*

$$C_{i,j} = \begin{cases} 0, & \text{if } j \notin N^i, \\ \sup_{s, a^i, a^j} \sup_{s, a^i, a'^j} \text{TV}\left(P\left(\cdot | s, a^i, a^j\right), P\left(\cdot | s, a^i, a'^j\right)\right), & \text{if } j \in N^i, j \neq i, \\ \sup_{s, a^i} \sup_{s, a'^i} \text{TV}\left(P\left(\cdot | s, a^i\right), P\left(\cdot | s, a'^i\right)\right), & \text{if } j = i, \end{cases}$$

*where $\text{TV}\left(\cdot, \cdot\right)$ is the total variation distance between two distributions. Further, define the DRI $Q$-function as*

$$\tilde{Q}_\theta^i\left(s, a^i\right) = \sum_{t=0}^{\infty} \mathbb{E}_{\boldsymbol{a}^{-N^i} \sim \boldsymbol{\pi}^{-N^i}, s \sim P}\left[\tilde{r}^i(s, \boldsymbol{a}) - \bar{\mu}_\theta^{N^i} | s_0 = s, a_0^i = a^i\right],$$

where $\tilde{r}^i$ follows Eqn. (2). If for all $i \in \mathcal{N}, \sum_{j=1}^N C_{ij} \leq \rho < 1$ and $\left|r^i(s, \boldsymbol{a})\right| \leq \bar{r}, \forall (s, \boldsymbol{a}) \in \mathcal{S} \times \mathcal{A}$, then the $\left(\frac{\bar{r}}{1-\rho}, \rho\right)$ exponential decay property holds, i.e.,

$$\left|\tilde{Q}_\theta^i\left(s, a^i, \boldsymbol{a}^{N-i}\right) - \tilde{Q}_\theta^i\left(s, a^i, \boldsymbol{a'}^{N-i}\right)\right| \leq \frac{\bar{r}}{1-\rho}\rho^2$$

holds for any $i \in \mathcal{N}, s \in \mathcal{S}, a^i \in \mathcal{A}^i, \boldsymbol{a}^{N-i}, \boldsymbol{a'}^{N-i} \in \mathcal{A}^{N-i}$.

*Proof.* The DRI method of Eqn. (2) reshapes original reward $r^i$ to $\tilde{r}^i$ and changes distribution of states and actions. Two conditions that ensures the exponential decay property of original $Q$-functions still establish for the reshaped $\tilde{Q}$, i.e., $\left|\tilde{r}^i(s, \boldsymbol{a})\right| \leq \bar{r}$ and $\sum_{j=1}^N C_{ij} \leq \rho < 1$ hold. From the proof of Theorem 1 in Qu et al. (2020), our Lemma 1 hold.

In Lemma 1, $C_{ij}$ is the maximum possible of the distribution of next state as a result of a change in node $j$'th current actions. The "change" is measured in total-variation distance, it can be interpreted as the strength of node $j$'s interaction with node $i$. The Lemma gives constraints on multi-agent networks that require the interaction strength between each pair of agents to be small enough and each agent to have a small enough neighborhood. It ensures the truncated $Q$-function $\tilde{Q}_\theta^i(s, a^i)$ is a good approximation of the global one $\tilde{Q}_\theta^i(s, \boldsymbol{a})$. Numerical experiments in Qu et al. (2020) show the broad establishment of the exponential decay property, and the empirical results in our paper demonstrate its practicability. In the following, standard assumptions in reinforcement learning are listed as the basis of theoretical convergence. The first one is that the rewards are bounded.

**Assumption 1.** For all $i$, and $s \in \mathcal{S}, \boldsymbol{a} \in \mathcal{A}^1 \times \cdots \times \mathcal{A}^N$, we have $0 \leq r^i(s, \boldsymbol{a}) \leq \bar{r}$.

The next is the condition on the step sizes for the two-time scale algorithm.

**Assumption 2.** The positive step size $\alpha_t, \beta_t$ are deterministic, non-increasing, square summable but not summable, i.e. $\sum_{t=0}^\infty \alpha_t = \sum_{t=0}^\infty \beta_t = \infty, \sum_{t=0}^\infty (\alpha_t^2 + \beta_t^2) < \infty$, and satisfy $\sum_{t=0}^\infty \left(\frac{\beta_t}{\alpha_t}\right)^d < \infty$ for some $d > 0$.

The next one is the uniformly ergodic of the underlying MDP.

**Assumption 3.** Under any policy $\theta = \left(\theta^1, \cdots, \theta^N\right)$, the induced Markov chain over the state-action space $\mathcal{S} \times \mathcal{A}$ is ergodic with stationary distribution $\pi^\theta$. Further, (a) For all $(s, a) \in \mathcal{S} \times \mathcal{A}$, $\pi(a|s) \geq \sigma$ for some $\sigma > 0$. (b) $\left\|P^\theta - \mathbf{1}\left(\pi^\theta\right)^T\right\|_{D^\theta} \leq \mu_D$ for some $\mu_D \in (0, 1)$, where $D^\theta = \mathrm{diag}\left(\pi^\theta\right) \in \mathbb{R}^{\mathcal{S} \times \mathcal{A} \times \mathcal{S} \times \mathcal{A}}$ and $\|\cdot\|_{D^\theta}$ is the weighted Euclidean norm $\|x\|_{D^\theta} = \sqrt{x^T D^\theta x}$ for vectors $x \in \mathbb{R}^{\mathcal{S} \times \mathcal{A} \times \mathcal{S} \times \mathcal{A}}$, and the corresponding induced norm for matrices.

The following is that the gradient is bounded and is Lipschitz continuous.

**Assumption 4.** For each $i, a_i, s$, $\left\|\nabla_{\theta^i} \log \pi_{\theta^i}\left(a^i|s\right)\right\| \leq L_i$. Let $L := \sqrt{\sum_{i \in \mathcal{N}} L_i^2}$. Further, $\nabla_{\theta^i} \log \pi_{\theta^i}\left(a^i|s\right)$ is $L'$-Lipschitz continuous in $\theta^i$, and $\nabla J(\theta)$ is $L'$-Lipschitz continuous in $\theta$.

Moreover, following the analysis path in Hu & Wellman (2003), our convergence result requires that the (original) stage games encountered during learning have global optima, or alternatively, that they all have saddle points.

**Assumption 5.** Every stage game $\left(Q_t^1, \cdots, Q_t^N\right)$, for all $t$ and $s$, has a global optimal point or a saddle point.

It would be unusual for stage games during learning to maintain adherence to Assumption 5. Nevertheless, the experiments in Hu & Wellman (2003) and in our paper practically show that convergence is not necessarily so sensitive to the properties of the stage games during learning. This condition established for convergence proof may have the potential to relax, which is beyond the scope of this paper. Finally, we obtain the following convergence proposition.

**Proposition 1.** *For an $N$-player stochastic game, Lemma 1 and Assumption 1-5 hold, the sequence* $\tilde{Q}_t = \left( \tilde{Q}_t^1, \cdots, \tilde{Q}_t^N \right)$, *updated by*

$$\mu_{t+1}^i = (1 - \alpha_t)\, \mu_t^i + \alpha_t \cdot r^i(s_t, \boldsymbol{a}_t),$$

$$\tilde{Q}_{t+1}^i\big(s_t, a_t^i\big) = (1 - \alpha_t)\, \tilde{Q}_t^i\big(s_t, a_t^i\big) + \alpha_t \cdot \left( \tilde{r}^i(s_t, \boldsymbol{a}_t) - \bar{\mu}_t^{N^i} + \tilde{Q}_t^i\big(s_{t+1}, a_{t+1}^i\big) \right),$$

$$\theta_{t+1}^i = \theta_t^i + \beta_t \cdot \tilde{Q}_t^i\big(s_t, a_t^i\big) \cdot \nabla_{\theta^i} \log \pi_{\theta_t^i}\big(a_t^i | s_t\big),$$

*where $\tilde{r}^i$ follows Eqn. (2), converges to an approximated Nash equilibrium point $\left( \pi_*^1, \cdots, \pi_*^N \right)$ with an asymptotic bound as $\lim\limits_{t \to \infty} \inf \|\nabla J(\theta_t)\| \le L \frac{\bar{r}\rho^2}{(1-\mu_D)(1-\rho)}$, and the $\tilde{Q}_* = \left( \tilde{Q}_*^1, \cdots, \tilde{Q}_*^N \right)$ constitutes a Nash Q-value.*

From Proposition 1, the DRI-based Actor-Critic algorithm can find an approximated stationary point with gradient size $O(\rho^2)$. Utilizing the game theory in Hu & Wellman (2003) and the stochastic approximation in Qu et al. (2020), Proposition 1 can be proved through the analogous procedure, that we omit here.

**Remark 2.** Each average reward tracker $\mu_t^i$ is updated by the respective original rewards $r_{t+1}^i$ rather than the reshaped one $\tilde{r}_{t+1}^i$. This is critical for agents to restructure the original global objective of Eqn. (1) by $\bar{\mu}^{N^i} = \frac{1}{N^i} \sum_{j \in \mathcal{N}^i} \mu_{\boldsymbol{\pi}}^j$.

## B ADDITIONAL EXPERIMENTAL RESULTS

### B.1 A THREE-PLAYER MATRIX GAME

| | | Scooter | | | | | |
|---|---|---|---|---|---|---|---|
| | | Henry's room | | | Cornfield | | |
| | | **Henry** | | | **Henry** | | |
| | | Henry's room | Cornfield | Classroom | Henry's room | Cornfield | Classroom |
| **Gus** | Henry's room | (-7,-7,10) | (6,-2,6) | (6,-2,6) | (9,9,-4) | (0,6,6) | (0,2,2) |
| | Cornfield | (-1,5,6) | (6,6,0) | (-2,-2,0) | (6,0,6) | (7,7,6) | (6,-2,6) |
| | Classroom | (-1,5,6) | (-1,-2,0) | (5,-2,0) | (-1,0,-2) | (0,6,6) | (5,-2,-4) |

(a) Payoffs of the three-player matrix game

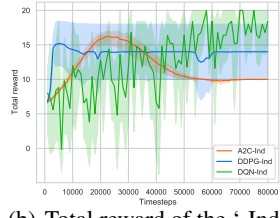 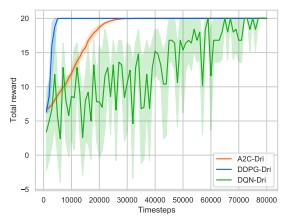

(b) Total reward of the '-Ind's    (c) Total reward of the '-Dri's

Figure 6: Results of the three-player matrix game.

A 3-player matrix game is provided in Fig. 6(a) from Matthew Rousu's Game Theory course. [2] It is a question about meeting during COVID. Two students, Gus and Henry, want to hang out together on a Friday night. They are deciding where to go to drink some juices. They also know Scooter and don't mind being around him, but they are a bit leery as Scooter is suspected to be a member of the Secret COVID Policy Force. They think Scooter might bust students for COVID violations if multiple people try to gather in an unapproved indoor space together. Gus and Henry could choose to

---

[2] www.youtube.com/watch?v=DdLqwsMKjMc.

go in Henry's room, in an empty classroom, or in the cornfield. Scooter doesn't know the classroom is an option, so will either visit Henry's room or the cornfield.

To solve this question, the communication network of the '-Dri's is set to be fully connected, and different '-Ind' and '-Dri' algorithms are performed. The total reward curves in Fig. 6 show that DRIMA enables three agents to learn the destination strategies with the optimal global reward, i.e., to meet in the cornfield valued at (7,7,6). The '-Ind's may converge to the global optimum or the inferior ones, both are Nash equilibriums. In other words, in general games with multiple equilibria, the strategy convergence result of decentralized MARL is sensitive to the initial condition and lacks collaboration to approach the global optimum. The DRI method enables agents to collaboratively achieve better solutions, in this case, the global optimal one.

### B.2 MULTI-AGENT PARTICLE ENVIRONMENT

#### B.2.1 SADDLE SITUATIONS IN MPE TASKS

Table 2: Payoff matrix for Cooperative Hunting, where (5,0) is the saddle point and (15,-5) is the global optimum.

| | | Intercepter | |
|---|---|---|---|
| | | Approach | Avoidance |
| **Attacker** | Approach | 15, -5 | 5, 0 |
| | Avoidance | 0, -5 | 0, 0 |

Table 3: Payoff matrix for Cooperative Collection, where (10,5) is the saddle point and (20,0) is the global optimum.

| | | Purple | |
|---|---|---|---|
| | | Approach | Leave |
| **Green** | Approach | 10, 5 | 20, 0 |
| | Leave | 0, 10 | 0, 0 |

For the continuous scenario defined with the continuous reward function, we can simplify and discretize agents' rewards as the matrix game to facilitate analysis while not affecting the validity. Table 2 denotes a saddle situation agents in the hunting task would encounter. When the attacker and interceptor learn independently, the former (with firepower) would actively approach and attack prey, while the latter (without firepower) would approach but avoid getting too close to the target and receiving collision damage. This could lead to two of policies to the saddle point (valued at (5, 0)). To get the optimal point(valued at (15, -5)), agents are required to learn to choose the "Approach" cooperatively. Table 3 is set to denote a saddle issue in the collection task, where two types of agents (Green and Purple) surround a Green food. In this case, the lack of cooperation agents would separately attempt to occupy the food (which one would succeed is stochastic) leading to the saddle equilibrium (valued at(10,5)). To achieve the global optimum (valued at(20,0)), the Purple agent needs to learn to leave this food occupied by the Green agent to obtain the highest type-matching reward. The actual situation of these two tasks is more complex since more agents are involved and rewards are dense, highly effective collaborative learning is required to achieve the ideal overall performance.

#### B.2.2 COOPERATIVE HUNTING

The group strategies of success and failure in Cooperative Hunting I are demonstrated in Fig. 7(a) and (b) respectively. As can be seen in Fig. 7(a), the two interceptors (greens) of the predators learned the cooperative strategy that surrounding the prey from outside the field of view and blocking its escape route to help the attacker(blue) hit the prey. Fig. 7(b), however, shows that three predators chased the prey from the same direction without forming an encirclement. Since the prey is faster, it can easily escape from blank directions. The DRI method enables multiple agents to learn collaborative strategies successfully.

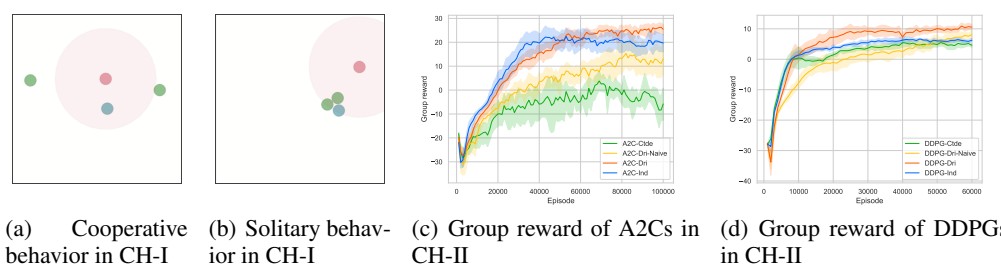

| (a) Cooperative behavior in CH-I | (b) Solitary behavior in CH-I | (c) Group reward of A2Cs in CH-II | (d) Group reward of DDPGs in CH-II |

Figure 7: Results in Cooperative Hunting.

Fig. 7(c) and (d) are the results of Cooperative Hunting II, in which 2 attackers and 2 interceptors hunt 2 prey. It can be seen that the performance of the centralized algorithm '-Ctde' decreases significantly as the number of agents increases, and the global reward can no longer accurately discriminate the situation of strategy combination. The '-Ind's are more likely to catch prey and obtain positive rewards due to the number of agents in the field increasing. The '-Dri's receive the highest reward and learn the best overall strategies through effective local information interactions, which are not affected by the growth number of agents.

### B.2.3 RESULTS OF A2C IN COOPERATIVE COLLECTION

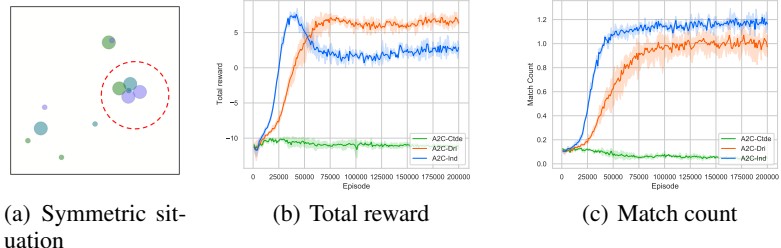

| (a) Symmetric situation | (b) Total reward | (c) Match count |

Figure 8: Results of A2Cs in Cooperative Collection.

In the Cooperative Collection, where 6 agents randomly search and occupy 6 foods, the stochastic policy algorithm fails to learn a good strategy combination. The situation A2C faced is pictured in Fig. 8(a), the total reward and match count are shown in Fig. 8(b) and (c). From Fig. 8(a), one food happens to be within the view of the four agents, and its position is considered to be the closest to each agent. These agents simultaneously approach and attempt to occupy that food. Since the space is continuous and each agent adopts a highly random exploration strategy, no agent has the advantage in occupying the target, i.e., no one of them successfully settles in the target without being pushed out by others. Agents are at an impasse. In contrast with A2C, multiple agents using DDPG are easy to break this deadlock due to their more deterministic and less stochastic policies. Theoretically, this situation constitutes a symmetric game in which the identities of the players can be changed without changing the payoff to the strategies. In the cases that coexist symmetric games, continuous space, and stochastic policy, the independent and simultaneous policy MARL will face challenges. We leave this problem as future work.

## C   PSEUDO-CODE OF DRIMA

Algorithm 1 represents the pseudocode of the Differential Reward Interaction-based Actor-Critic, where the individual $V$-function and policy are parameterized by $\omega^i$ and $\theta^i$ respectively. Other algorithm variants of RL can be obtained by analogy.

972
973
974
975
976
977
978
979
980
981
982
983
984
985
986
987
988
989
990
991
992
993
994
995
996
997
998
999
1000
1001
1002
1003
1004
1005
1006
1007
1008
1009
1010
1011
1012
1013
1014
1015
1016
1017
1018
1019
1020
1021
1022
1023
1024
1025

---

**Algorithm 1** Differential reward interaction-based Actor-Critic

---

1: **Input:** Initial value of parameters $\mu_0^i, \omega_0^i, \theta_0^i, \forall i \in \mathcal{N}$, the initial state $s_0$, and step-sizes $\{\alpha_t\}_{t \geq 0}$ and $\{\beta_t\}_{t \geq 0}$.
   Initialize the iteration counter $t \leftarrow 0$.
2: **repeat**
3:     **for** all $i \in \mathcal{N}$ **do**
4:         Each agent $i$ executes action $a_t^i \sim \pi_{\theta_t^i}^i (\cdot | s_t)$.
5:         Observe state $s_{t+1}$, and reward $r_t^i = r^i (s_t, a_t^i)$.
6:         Update $\mu_{t+1}^i \leftarrow \mu_t^i + \alpha_t \cdot (r_t^i - \mu_t^i)$.
7:     **end for**
8:     **for** all $i \in \mathcal{N}$ **do**
9:         Receive $\mu_t^j, r_t^j$ from neighbors $j \in \mathcal{N}^i$.
10:        Calculate $\tilde{r}_t^i - \bar{\mu}_t^{N^i}$ according to Eqn. 2.
11:        Update $\delta_t^i \leftarrow \tilde{r}_t^i - \bar{\mu}_t^{N^i} + V_{\omega_t^i}^i (s_{t+1}) - V_{\omega_t^i}^i (s_t)$.
12:        **Critic step:** $\omega_{t+1}^i \leftarrow \omega_t^i + \alpha_t \cdot \delta_t^i \cdot \nabla_{\omega^i} V_{\omega_t^i}^i (s_t)$.
13:        **Actor step:** $\theta_{t+1}^i \leftarrow \theta_t^i + \beta_t \cdot \delta_t^i \cdot \nabla_{\theta^i} \log \pi_{\theta_t^i}^i (a_t^i | s_t)$.
14:     **end for**
15:     Update the iteration counter $t \leftarrow t + 1$.
16: **until Convergence**

---

