# OpenReview forum: "DRIMA: Differential Reward Interaction for Cooperative Multi-Agent Reinforcement Learning"
_ICLR.cc/2025/Conference — Submitted to ICLR 2025_

### Official Review · Reviewer_4rDm · 2024-10-26

**Soundness:** 3
**Presentation:** 3
**Contribution:** 3
**Rating:** 6
**Confidence:** 4

**Summary:**

This paper follows the distributed training with decentralized execution (DTDE) training paradigm and focuses on the cooperation between multiple distributed agents. The authors argue that the conflicting payoffs of different agents would emphasize the game between agents and hinder cooperative behaviors. To mitigate this, they proposed a conflict-triggered differential reward interaction (DRI) method to reconstruct the individual rewards. The proposed method, DRIMA, is tested on matrix games, MPE, and SMAC to verify its effect.

**Strengths:**

1. This article provides a detailed description of the use of DRIMA in the context of specific tasks from easy (Prisoner's dilemma) to hard (general Markov games). The pseudo-code in the appendix also makes the proposed method easier to follow.
2. The authors have proved the convergence of learning with the differential reward, which makes DRIMA technically solid.
3. Experiment and theory corroborate each other very well in DRIMA. The use of DRIMA effectively helps the raw algorithms jump out NE and reach the global optimum in matrix games.

**Weaknesses:**

1. The performance of DRIMA in SMAC is not significant. Even with the inclusion of communication between agents, DRIMA is still inferior to CTDE in some scenarios. It makes me concerned about the value of the proposed method in real-world tasks.
2. It seems DRIMA often exhibits a slower start in experiments. This is detrimental to sample efficiency, and I suggest the authors give further explanation or improvements.

**Questions:**

1. When tested on SMAC, how do the authors define the individual rewards of the agents?
2. See Weaknesses 2.

---

> ### Author Response · Authors · 2024-11-29
> **Response to Reviewer**
>
> The authors would like to sincerely thank the reviewer for the careful reading and positive comments on our paper. The following are responses to the Weaknesses and Questions.
>
> To the **Weaknesses**:
> 1. Please review the updated PDF of our paper, more results of SMAC scenarios have been added. The wide range of scenarios shows that DRIMA exhibited relatively slower learning efficiency at the early stage in some scenarios, but it did not impair learning results when run until convergence.
> In fact, distributed learning approaches compared with independent and centralized ways inherently possess slower learning rate, since the distributed communication between agents aim to reconstruct the useful global information which is directly available to centralized algorithms and is not considered by independent algorithms.
> However, independent (or centralized) and fast learning efficiency do not guarantee the overall collaborative effectiveness and robustness of the strategy combinations.
> DRIMA is suitable for real-world multi-agent tasks, where centralized payoff information is difficult or expensive to obtain, or that demands system robustness.
> 2. We give an explanation in the **1**st reply, and from our experiments, the relatively slower sample efficiency of DRIMA had no detrimental impact on the final convergence results.
>
> To the **Questions**:
> 1. Following the reward construction of SMAC in MAPPO (Yu et al., 2022), the individual rewards are set as three parts, the first one is an Enemy Group reward that sums up all enemies' instant health value which allows the agent to grasp the overall battle progress, the second one is a reward relating to an Ally agent's own health, and the final part is a win reward given to all Ally agents if win the battle.
>
> We hope that the above responses have addressed your questions. If you have any further confusion, we'd be glad to discuss them with you.

---

> ### Comment · Reviewer_4rDm · 2024-11-30
>
> The reviewer thanks the authors for their efforts. After reading the rebuttal and the updated draft, I still have some questions about the experiments. The authors assert that the slower learning efficiency at the early stage of training is due to the absence of useful global information, and DRIMA is suitable for tasks in which centralized information is unavailable. But in experiments, global information like all enemies' instant health is still used for reward construction. Would this contradict the distributed setting? Is DRIMA only allowed to receive data from global but not to send messages? If so, why not provide global information for each agent in SMAC experiments to see if the learning efficiency can be improved?

---

> > ### Author Response · Authors · 2024-12-02
> > **Official Comment by Authors**
> >
> > The authors thank the reviewer for the insightful question.
> > As for the SMAC scenario, setting individual Ally agent's reward merely relates to local enemies (within its view range or being attacked by it) will cause a **time-delay reward** issue.
> > For instance, Ally agent $i$ attacks an enemy when it is in good health and gets a relatively small immediate reward (the enemy has a relatively small health loss), while Ally agent $j$ gets more reward by attacking the enemy when it is in poor health.
> > In this case, part of agent $j$'s instant reward is a delayed reward of agent $i$, since agent $i$'s early attack causes the enemy's health to deteriorate, but this part is not fed back to agent $i$.
> > The time-delayed reward across multiple agents is still an issue that needs further investigation in MARL, to avoid this issue here, agents are coarsely set to receive the entire enemies' instant health reward.
> >
> > On this basis, we set individual Ally agents to receive a distinct reward relating to their own health status.
> > DRIMA still works between multiple learning agents with distinctive rewards and differential reward interactions to reconstruct a global Ally situation.
> > As for providing a global reward for each agent in SMAC the reviewer mentioned, the learning curves are plotted as 'PPO-Ctde', in which each agent receives a total Ally health reward instead.
> > Experimental results show that 'PPO-Dri' generally performs relatively slower rate but comparable convergence results with 'PPO-Ctde', while obtaining an outstanding performance in the '5m_6m' task.
> > This indicates the typical trade-off problem between an efficient centralized approach and a relatively slower distributed way but with potentially enhanced collaboration strategies.

---

### Official Review · Reviewer_H3gs · 2024-11-03

**Soundness:** 2
**Presentation:** 3
**Contribution:** 2
**Rating:** 5
**Confidence:** 3

**Summary:**

The paper proposes a conflict-triggered differential reward interaction (DRI) method to identify and reshape conflicting payoffs between agents. The experiments conducted on matrix games and continuous space tasks MPE demonstrate the effectiveness of the algorithm.

**Strengths:**

1. The paper is well-motivated overall.
2. The paper proposes deals with saddle equilibriums based on conflict-triggered differential rewards.

**Weaknesses:**

1. The proposed algorithm lacks experimental comparisons with recent works on MARL cooperation.
2. The current Figure 2 is not very comprehensible for understanding the conflict-triggered mechanism of DRI. The boundaries between the various colored surfaces are not easily distinguishable.
3. Including experimental results from more complex SMAC scenarios in the main paper would better demonstrate the applicability of the proposed method.

**Questions:**

1. As the method exploit the MFT concept, could it be applied to large-scale multi-agent environments?
2. I am wondering whether conflict-triggered differential rewards might hinder early exploration.

---

> ### Author Response · Authors · 2024-11-29
> **Response to Reviewer**
>
> The authors would like to gratefully acknowledge the reviewer for the valuable time during this review. We appreciate the strengths of your comment on our paper, and the following are our responses to your remaining concerns.
>
> To the **Weaknesses**:
> 1. The current representative MARL cooperation often has limited generality.
> Such as **COMA** (Foerster et al., 2018) takes joint action $\mathbf{a}$ to calculate global-level advantage value $A^{a^i}\left( s, \mathbf{a} \right) $, possessing limited scalability.
> **VDN** (Sunehag et al., 2017) and **QMIX** (Rashid et al., 2020) respectively designs the NN structure which inputs N agents' local action-value function $Q^i\left( o^i, a^i \right) \left( i=1,...,N \right) $ and outputs the total joint value function $Q^{tot}\left( \mathbf{o}, \mathbf{a} \right) $.
> The two approaches are well applicable for the discrete action space, which can be represented as the one-hot vector and is discriminative enough as the NN inputs, to ensure the effective join action-value functions learning, while not quite fitting in the continuous action space.
> **MFRL** (Yang et al., 2018) learned through independent reward that is incapable of dealing with saddle equilibriums.
> Our approach is applicable for both value-based (e.g. DQN) and policy-based (e.g. A2C, DDPG, and PPO) RL algorithms, and for both discrete and continuous space environments.
> For discrete action space, we adopted **MAPPO** (Yu et al., 2022) as SOTA; for continuous action space, we selected the widely used benchmark **MADDPG** (Lowe et al., 2017).
> To make the experimental results straight and concise, we chose the 'Independent' and 'Centralized' forms of those algorithms as comparisons.
> 2. Thank you for this suggestion, we have refined Figure 2(a) to help identify various colored surfaces, and improve the related description under it for enhanced comprehensibility, please refer to the highlighted content in the updated PDF.
> 3. Thank you for the constructive suggestion, we have included the SMAC results in the main paper of the revision. Additionally, more results of battle scenarios have been included to improve the completeness.
>
> To the **Questions**:
> 1.  DRIMA is based on local information exchange, the scale of this approach does not increase with the number of agents in a system.
> Therefore, it can be equipped on almost every existing MARL, including MFRL (Yang et al., 2018), for saddle equilibrium avoidance and collaboration enhancement in large-scale environments.
> 2. The wide range of experiments in our paper shows that conflict-triggered DRI exhibited relatively slower learning efficiency at the early stage in some scenarios, but it did not impair convergence results.
> In fact, distributed learning algorithms compared with independent and centralized ones inherently possess slower learning rate, since the distributed communication between agents aim to reconstruct the useful global information which is directly available to centralized algorithms and is not considered by independent algorithms.
> However, independent (or centralized) and fast learning efficiency do not guarantee the overall collaborative effectiveness and robustness of the strategy combinations. This is a trade-off problem and an ongoing challenge in the distributed learning study.
>
> We hope the above response has addressed your main concern about our paper, and we'd be glad to discuss any further questions you may have. The authors would greatly appreciate it if you could consider increasing your rating for the paper after reviewing the above response and the revised PDF.

---

> ### Author Response · Authors · 2024-12-02
> **Official Comment by Authors**
>
> The authors thank the reviewer for the valuable time and suggestions proposed to improve our work.
> If all your concerns have been addressed, we kindly ask you to consider raising your rating.
> If you still have any doubts or reservations about our work, please feel free to give us further feedback, we are more than willing to engage in further discussion with you.

---

### Official Review · Reviewer_F57D · 2024-11-03

**Soundness:** 4
**Presentation:** 2
**Contribution:** 3
**Rating:** 6
**Confidence:** 4

**Summary:**

Although the Centralized training with decentralized execution (CTDE) framework is currently popular, it faces challenges in terms of scalability and handling complex tasks. This paper introduces a new approach called differential reward interaction (DRI) with conflict-triggered, which enables agents to achieve a unified overall goal by exchanging local information with each other. This method also overcomes the issue of saddle equilibrium in policy combination, a common challenge in game theory and optimization. The effectiveness and superiority of the proposed method, DRIMA, are demonstrated through its application in matrix games, Multi-Agent Particle Environment (MPE) scenarios, and Star-CraftⅡ.

**Strengths:**

1.	The overall structure of the paper is complete, the content is sufficient, and the experimental evidence is persuasive.
2.	The paper proposes that if personal rewards conflict with neighbor rewards, personal rewards should be reshaped as the neighbor-averaged reward to maintain the group consistency. This idea is similar to Reward Centering or Advantage Functions.

**Weaknesses:**

1.	The paper does not clearly define "differential rewards" or "conflicts".
2.	The purpose of Equation 2 is not intuitively explained. Providing a straightforward example would greatly help in understanding it.
3.	The relationship between Equation 2 and the differential action-value function is not explained.
4.	In Section 2.2, $\mu_\pi^i$ in the differential action-value function represents the average reward per step for agent i. However, in Section 3.2, $\bar\mu_\pi$ represents the average of the average rewards per step for the agent itself. Is there a contradiction between the two? So, I think the formula in section 2.2 may have been written incorrectly.
5.	Around lines 244-247, should not $\beta_t$ in Equation 4 be the step size of the Actor?
6.	The example in Figure 2 is somewhat abstract. Could you explain in more detail the issue of gradient directions under different views?

**Questions:**

1.	Did you draw on the idea of the advantage function? If so, where does the specificity of your method lie? What are the advantages compared to methods using the advantage function?
2.	Did you draw on the concept of Reward Centering? If so, what is the specificity of your method?
3.	Some questions about the Weakness section.

---

> ### Author Response · Authors · 2024-11-29
> **Response to Reviewer**
>
> The authors would like to sincerely thank the reviewer for the careful reading and positive comments on our paper. The following are responses to the Weaknesses and Questions.
>
> To the **Weaknesses**:
> 1. The "differential reward" of an agent is $r^i-\mu ^i \left( i=1,...,N \right) $, and the "conflict" is the opposite sign between $r^i-\bar{\mu}^{N^i}$ and $\bar{r}^{N^{-i}}-\bar{\mu}^{N^i}$. We refined **Section 2.2** and **3.1** with more clear definitions, please check the updated PDF.
> 2. We described the purpose of Equation 2 in the paragraph above the title of Section 3.1, and the intuition of this approach is derived from the "differential reward" concept in traditional (single-agent) RL, which was described in the last paragraph of Section 3.1.
> Thank you for this comment, we have refined the related description in **Section 3.1** to be more clear in the revision.
> 3. As the ordinary definition of the differential value function in Section 2.2, the reshaped differential value function by Equation 2 can be obtained by analogy.
> Its formal definition was given in Lemma 1 and its updated rule was in Proposition 1, which we put in the (Convergence Analysis) Appendix A.2 considering the space limit.
> 4. The formula in Section 2.2 is correct, where $\mu_{\pi}^{i}$ represents the average reward of agent $i$ before DRI, and $\bar{\mu}_{\pi}$ in Section 3.2 denotes the average of the neighboring average rewards after DRI.
> 5. The $\beta_t$ should be the Actor step-size, thank you for pointing out this typing mistake.
> 6. Thank you for this comment, we have refined the related description and Figure 2(a) in the revision, please refer to the highlighted content under Figure 2.
>
> To the **Questions**:
> 1. **One of** the differences between the advantage function ''$Q_t\left( s, a \right) -V_t\left( s \right) $'' and the differential reward ''$R_t\left( s, a \right) -\mu , \mu =\underset{T\rightarrow \infty}{\lim}\frac{1}{T}\sum_{t=0}^{T-1}{\mathbb{E} _{s\sim P, a\sim \pi}\left[ R_t\left( s,a \right) \right]}$'' is the former takes value function to compute, and in deep RL tasks the functions are approximated by NNs.
> In multi-agent scenarios, interactions using these approximations are subject to certain inaccuracies and inefficiencies, while the differential reward term is more precise.
> In fact, we had conducted experiments that combined the advantage function and our conflict-triggered idea, the results were in general inferior to the conflict-triggered DRI.
> **Another** advantage of DRI is that the average reward term ''$\mu$'' directly reflects the policy's expected reward that relates to the objective of the agent during learning, interactions utilizing it in the learning process help multiple agents keep goal consistency efficiently.
> Therefore, we believe the advantages of multiple dimensions work together that ensure the superiority of our approach.
> 2. The idea of our paper was derived from the *average reward* concept in the traditional (single-agent) reinforcement learning theory (Sutton & Barto, 2018).
> It is drawn from dynamic programming as one of the value function formulation methods in RL.
> The recent paper Reward Centering brings this concept back, restudy it with the Laurent-series decomposition theory, and gives it a new description as "Reward Centering".
> The main content of it is to explicate that the average reward concept plays an essential role in generally improving (single-agent) RL algorithms.
> Our study was accomplished before Reward Centering was published, and we hold a coincident view with it that "the simplest method can be quite effective", not only in single-agent learning, but we investigate it for cooperative multi-agent learning.
> (Note that all the compared algorithms in our experiments fairly utilized the average reward term.)
>
> We hope that the above responses have addressed your questions. If you have any further confusion, we'd be glad to discuss them with you.

---

### Official Review · Reviewer_pn7C · 2024-11-04

**Soundness:** 2
**Presentation:** 2
**Contribution:** 2
**Rating:** 5
**Confidence:** 2

**Summary:**

An important issue in DETD is the interactions among agents, which manifest in simultaneous decision-making and reward conflicts. To address these challenges, this paper introduces DRIMA, a differential reward interaction-based MARL method. Specifically, the authors model the information interactions among agents as an undirected graph network, where each node represents an agent and each edge represents the connections between them. Based on this, they propose an exchange reward rule based on the overall performance of the policy. They evaluate the quality of the team policy based on the differential reward of the current agent's action. The authors conduct a theoretical analysis of their method from the perspectives of two-player matrix games and multi-player Markov games. Their experiments focus on general-sum games, specifically validating the method's effectiveness in matrix games, MPE, and SMAC.

**Strengths:**

1.	The authors conducted a theoretical analysis of the convergence of their method.
2.	The authors used "Two-player matrix game" and "Multi-Player Markov Game" to analyze the effectiveness of their method.

**Weaknesses:**

1.	The interaction rule in line 164 lacks a formal definition, and $r^i-\bar{\mu}^{N^i}$  and $\bar{r}^{N^i}-\bar{\mu}^{N^i}$ lack more direct definitions, making it difficult to understand.
2.	The paper does not provide sufficient experiments in SMAC to demonstrate its effectiveness.

**Questions:**

1.	What’s the formal definition of “reward interaction”?
2.	In line 193, what do $r^i_t-\bar{\mu}_t^{N^i}$ and $\bar{r}_t^{N^{-i}} - \bar{\mu}_t^{N^i}$ represent? And how does the opposite sign between them “indicates the conflict contribution of action decision relating to the individual and other neighbors’to the current neighbor-joint policy”? As a key contribution of the paper, the authors need to carefully explain the meaning of each part in this section to help readers understand their ideas.

3.	In the authors' paper, each agent in the MAS is modeled with an individual reward. However, in the SMAC environment, each agent shares the same reward. How does the method in the paper address this situation?

4.	Based on Question 3, in many cooperative multi-agent systems, each agent shares a team reward, meaning that $r^i$ in Formula 2 is the same and equal to $\bar{r}^{N^i}$. Does the conflict-triggered differential reward interaction still apply in this case?

5.	In the MPE experiments, the method was not compared with QMIX. Please explain the choice of baselines.

---

> ### Author Response · Authors · 2024-11-29
> **Response to Reviewer**
>
> The authors would like to gratefully acknowledge the reviewer for the valuable time during this review. We appreciate the strengths of your comment on our article, and the following are our responses to your remaining concerns.
>
> To the **Weaknesses**:
> 1. We revised the definition to make it formal and more direct, and **Sections 2.2** and **3.1** are refined to improve the understandability, please check the highlighted content in our updated PDF.
> 2. We had done lots of experiments in SMAC, and we demonstrated the representative four results considering SMAC scenarios mainly belong to the four types (i.e., homogeneous and symmetrical, homogeneous and asymmetrical, heterogeneous and symmetrical, and heterogeneous and asymmetrical).
> Our algorithm and compared ones generally perform well in other scenarios.
> Considering your constructive suggestion, we have added more scenarios to improve the completeness, and we moved SMAC experiments from the Appendix to the main content of the revision.
>
> To the **Questions**:
> 1. The formal term is ''differential reward interaction'', it means that multiple agents interacting their differential rewards $r^i-\mu^i \left( i=1,...,N \right) $ during learning.
> The ''differential reward'' is one of the traditional formulations of the value function in (single-agent) RL (Sutton & Barto, 2018).
> We have refined this definition, please refer to Sections 2.2 and 3.1.
> 2. At the beginning of this paragraph, we firstly explained the differential reward term $r\left( s_t, a_t \right)-\mu_t$ for **single-agent** RL.
> Since $ \mu_t $ indicates the expected average reward $ \mu_\pi $ (defined in Section 2.2, under the differential action-value function), and from the definition, one can see that it reflects the overall assessment of its current policy.
> For **single-agent**, at training step $t$, the differential reward term $r\left( s_t, a_t \right) -\mu_t$ represents a contribution value to current policy by taking action $a_t$ at state $s_t$, and the sign of this term represents this contribution is positive or negative.
> Correspondingly, in **multi-agent** setting, the opposite sign between $\bar{r}^i\left( s_t, \mathbf{a}\_t \right)- \bar{\mu}\_{t}^{N^i}$ and $\bar{r}^{N^{-i}}\left( s_t, \mathbf{a}\_t \right) -\bar{\mu}\_{t}^{N^i}$ represents the conflict contribution of the action decisions relating to the individual $i$ and its other averaged neighbors $N^{-i}$, to the current neighbor-joint policy $\boldsymbol{\pi}_{t}^{N^i}$ (valued by $\bar{\mu}\_{t}^{N^i}$).
> When the opposite sign occurs at step $t$, agent's personal reward $r^i\left( s_t, \mathbf{a}_t \right) $ was reshaped as neighbor-averaged reward $\bar{r}^{N^i}\left( s_t, \mathbf{a}_t \right) $, to keep joint goal consistency.
> Reward consensus at critical learning steps can efficiently trade off the joint goal consistency and individual diversity, avoid saddle equilibriums, and ensure robust collaborative learning.
> Thank you for the comment, we have clarified these descriptions in **Section 3.1** of the revision.
> 3. In SMAC, all the learning agents of the Ally Group receive an Enemy Group reward that sums up all enemies' instant health value, it enables the agent to grasp the overall battle progress. Additionally, each individual receives their own health value as a reward for the 'Independent' and 'Drima' algorithms, while for the 'Ctde' algorithm, agents obtain a total Ally Group reward that summarizes all the members' health.
> 4. Based on the **3**rd reply, each agent receives different reward $r^i$ and composes different $\bar{r}^{N^i}$ respecting their communication range (we set it as view range).

---

> ### Author Response · Authors · 2024-11-29
> **Response to Reviewer**
>
> *(continued from the previous response)*
>
> 5. QMIX designed an extra NN structure, it takes N agents' local action-value function $Q^i\left( o^i, a^i \right) \left( i=1,...,N \right) $ as input and learns their weighted sum as the output that gain the total joint value function $Q^{tot}\left( \mathbf{o}, \mathbf{a} \right) $.
> This method is applicable for the discrete action space, since actions can be represented as one-hot vectors which are discriminative enough as the NN inputs to learn effective joint action-value functions.
> For the continuous action space scenario including MPE, QMIX is not well applicable.
> The paper of QMIX chose discrete action SMAC to conduct experiments, and the paper of MAPPO after it demonstrated that MAPPO overall outperforms QMIX, so we chose MAPPO as our baseline (labeled as PPO-Ctde in the experiments) in SMAC.
> For the continuous action MPE scenario, we choose the widely adopted MADDPG (labeled as DDPG-Ctde) as a baseline.
> The current cooperative learning approaches often have limited generality, our approach DRIMA is applicable for both value-based (DQN, etc.) and policy-based (A2C, DDPG, PPO, etc.) RL algorithms, and for both discrete and continuous space environments.
> To make the experimental results straight and concise, we chose the 'Independent' and 'Centralized' forms of those algorithms as comparisons, generally, DRIMA can be equipped on almost every existing MARL for saddle equilibrium avoidance and collaboration enhancement.
>
> We hope the above response has addressed your main concern about our paper, and we'd glad to discuss any further questions you may have. The authors would greatly appreciate it if you could consider increasing your rating for the paper after reviewing the above response and the revised manuscript.

---

> > ### Author Response · Authors · 2024-12-02
> > **Official Comment by Authors**
> >
> > The authors thank the reviewer for the valuable time and suggestions proposed to improve our work.
> > If all your concerns have been addressed, we kindly ask you to consider raising your rating.
> > If you still have any doubts or reservations about our work, please feel free to give us further feedback, we are more than willing to engage in further discussion with you.

---

### Meta-Review · Area_Chair_PNTj · 2024-12-21

**Metareview:**

This paper presents a differential reward interaction-based MARL method that addresses agent cooperation through conflict-triggered reward reshaping. While the technical approach shows promise in handling saddle equilibrium problems and demonstrates effectiveness in matrix games and MPE scenarios, reviewers identified several areas needing clarification and improvement. Key concerns included unclear definitions of "differential rewards" and "conflicts," limited experimental validation in SMAC environments, and questions about the method's applicability to shared-reward scenarios. The theoretical foundation and convergence analysis were well-received, though questions remained about learning efficiency and practical deployment. The authors are encouraged to improve this work in these aspects, particularly using more complex benchmarks.

**Additional Comments On Reviewer Discussion:**

The authors provided clearer definitions of key concepts, expanded SMAC experiments, and detailed explanations of the reward interaction mechanism. While some reviewers' concerns were adequately addressed, some reviewers maintained reservations about the method's practical applicability and sample efficiency.

---

### Decision · Program_Chairs · 2025-01-22

Reject